# Targeted Inhibitors of EGFR: Structure, Biology, Biomarkers, and Clinical Applications

**DOI:** 10.3390/cells13010047

**Published:** 2023-12-25

**Authors:** Nina Shaban, Dmitri Kamashev, Aleksandra Emelianova, Anton Buzdin

**Affiliations:** 1Shemyakin-Ovchinnikov Institute of Bioorganic Chemistry, Moscow 117997, Russia; dkamashev@gmail.com (D.K.); buzdin@oncobox.com (A.B.); 2Laboratory for Translational Genomic Bioinformatics, Moscow Institute of Physics and Technology, Dolgoprudny 141701, Russia; 3Institute of Personalized Oncology, I.M. Sechenov First Moscow State Medical University, Moscow 119991, Russia; 4World-Class Research Center “Digital Biodesign and Personalized Healthcare”, Sechenov First Moscow State Medical University, Moscow 119991, Russia; emelianova-a.g@yandex.ru; 5PathoBiology Group, European Organization for Research and Treatment of Cancer (EORTC), 1200 Brussels, Belgium

**Keywords:** epidermal growth factor receptor (EGFR), HER-targeted drugs, EGFR-targeting drugs, secondary resistance, EGFR mutations

## Abstract

Members of the EGFR family of tyrosine kinase receptors are major regulators of cellular proliferation, differentiation, and survival. In humans, abnormal activation of EGFR is associated with the development and progression of many cancer types, which makes it an attractive target for molecular-guided therapy. Two classes of EGFR-targeted cancer therapeutics include monoclonal antibodies (mAbs), which bind to the extracellular domain of EGFR, and tyrosine kinase inhibitors (TKIs), which mostly target the intracellular part of EGFR and inhibit its activity in molecular signaling. While EGFR-specific mAbs and three generations of TKIs have demonstrated clinical efficacy in various settings, molecular evolution of tumors leads to apparent and sometimes inevitable resistance to current therapeutics, which highlights the need for deeper research in this field. Here, we tried to provide a comprehensive and systematic overview of the rationale, molecular mechanisms, and clinical significance of the current EGFR-targeting drugs, highlighting potential candidate molecules in development. We summarized the underlying mechanisms of resistance and available personalized predictive approaches that may lead to improved efficacy of EGFR-targeted therapies. We also discuss recent developments and the use of specific therapeutic strategies, such as multi-targeting agents and combination therapies, for overcoming cancer resistance to EGFR-specific drugs.

## 1. EGF Receptor Protein Family

In humans, the EGF receptor family (ERBB/HER) consists of four structurally related receptor tyrosine kinases (RTKs) that regulate proliferative cell signaling and play pivotal roles in both normal physiology and proliferative diseases like cancer [1]. The four family members are EGFR/ErbB1/HER1, ErbB2/Neu/HER2, ErbB3/HER3, and ErbB4/HER4 proteins [2], which are encoded, respectively, by genes EGFR, ERBB2, ERBB3, and ERBB4 [3]. These genes are located on four different chromosomes, but their products share common structural organization, including an extracellular domain, lipophilic transmembrane region, intracellular domain with tyrosine kinase activity, and a carboxy-terminal region [4].

The ERBB/HER family members are expressed in epithelial, mesenchymal, and neuronal cells and in their cellular progenitors [5]. The family members play central roles in cell proliferation, survival, differentiation, adhesion, and migration. These molecules interconnect the inner and outer compartments of the cytoplasmic membrane and trigger the cellular responses to various external stimuli by transmitting the intracellular regulatory stimuli [6]. The activated ERBB/HER receptors form regulatory complexes in which components can enter the cytoplasm and promote downstream molecular pathways (Figure 1), including well-known oncogenic pathways of RAS-RAF-MEK-ERK and AKT-PI3K-mTOR signaling axes [7]. Furthermore, apart from dimerization, EGFR molecules can also form oligomers on the cell surface, both under the action of natural ligands or in their absence [8,9]. The phenomenon of EGFR oligomerization is thought to be important for intracellular signaling because it results in a tight organization of kinase-active molecules in a manner that is optimal for autophosphorylation in trans between adjacent dimers [10].

Several growth factors are known to be able to bind ERBB/HER receptors and activate them. These are the members of the epidermal growth factor (EGF) family, which are generally classified into three groups. Representatives of the first one bind only to EGFR, which includes EGF [11], transforming growth factor alpha (TGF-α) [12], epigen (EPG) [13], and amphiregulin (AR) [14]. The second group has dual specificity of receptor binding and includes betacellulin (BTC) [15], heparin-binding epidermal growth factor (HB-EGF) [16], and epiregulin (EPR) [17]. The third group consists of neuregulins (NRG) and forms two subgroups depending on their ability to bind both HER3 and HER4 (NRG1 and NRG2 [18]) or only HER4 (NRG3 and NRG4 [19,20]) (Figure 2a).

The inactivated forms of EGFR, HER3, and HER4 receptors exist in a pre-dimerized state. In turn, binding of the specific ligand causes rearrangement of the respective subunit of the receptor by turning the transmembrane domains. Activation leads to internalization of the receptor and trafficking to the early endosomal compartment of the cell. Next, endocytosis sorting occurs, whereby the receptor is either transported to the lysosome for further degradation or recycled to occupy a place in the cell membrane [21]. The family ligands affect receptor internalization in a different manner: upon EGF binding, the majority, but not all EGFRs, are continuously ubiquitinated and transported to lysosomes. HB-EGF and BTC also behave the same way. On the other hand, when subjected to the low pH of endosomes, TGF-α, EPR, and AR quickly separate from the receptor, which leads to de-ubiquitination of the receptor and its subsequent recycling to the plasma membrane (Figure 2b) [22].

In contrast with other HER family members, none of the ligands bind to HER2 [23]; it always exists in the dimerized state and acts as a preferred partner for heterodimerization with the other three ERBB/HER family members [24]. Also, HER2-containing heterodimers are characterized by higher affinity and broader ligand specificity than other heterodimeric ERBB/HER receptor complexes due to the slower dissociation rates of growth factors [25]. There was a controversy regarding HER3 pertaining to its kinase activity, and initially, it was posited that HER3 lacked kinase activity due to the absence of requisite residues [26]. Later reports, however, have suggested that HER3 does possess tyrosine kinase activity in some degree [27,28].

Ligand binding induces the formation of homo- and heterodimers by the ERBB/HER receptors and activates their internal kinase domain, which leads to the cross-phosphorylation of the tyrosine residues in the cytoplasmic tail. In turn, these phosphorylated tyrosine residues serve as the binding sites for a number of downstream regulatory proteins that activate intracellular signaling, which, in the case of EGFR activation, leads to proliferation and evasion of apoptosis [1,29].

## 2. EGFR Role in Cancer

Mutations and cases of overexpression of EGFR are especially frequently found in carcinomas and glioblastomas, tumors of epithelial and glial origin, respectively [30,31]. Worldwide, carcinomas are the most common type of cancer [32]. Overexpression of EGFR has been reported and implicated in the pathogenesis of many human malignancies, including head and neck [33], lung [34], breast [35], pancreatic [36], and colon cancer [37]. The EGFR-positive status of the tumor often correlates with poor prognosis and outcome, as it is beneficial for cancer cell proliferation [7,38]. EGFR overexpression was also shown to be associated with melanoma progression and promoted invasiveness and metastasis in this tumor type [39].

According to various clinical investigations, in non-small-cell lung cancer (NSCLC), EGFR-targeted therapy with gefitinib or erlotinib benefit was limited to NSCLC tumors bearing activating mutations of EGFR [40]. Still, gefitinib or erlotinib therapy can be efficient even in NSCLC patients with wild-type EGFR, where the predictive biomarkers remain unknown [41], whereas the objective response rate here does not exceed 21% [42].

Mutations in the tyrosine kinase domain of EGFR were found in the majority of tumors that exhibited a positive response to treatment with EGFR-specific TKIs (Figure 3a in green) [43]. In some reports, the frequency of EGFR-activating mutations has strong ethnical specificity and varies by region, being as high as 46% in Asia versus only 8% in the Americas [44]. The two most common mutations of EGFR in NSCLC represent about 85–90% of all EGFR mutations [45]. The first one is a deletion of EGFR exon 19 (*del747–750*), which eliminates the leucine-arginine-glutamate-alanine motif in the tyrosine kinase domain of EGFR (*LREA deletion*), and the second one (*L858R*) is a thymine-to-guanine transversion, which results in the replacement of leucine with arginine in exon 21 codon 858 [46,47]. The third most frequent type of EGFR mutations in NSCLC is exon 20 insertions (ex20ins), which constitute 9% [48]–12% [49] of all EGFR mutations. In contrast to the other above mentioned mutations, Ex20ins is associated with poor response to treatment with TKIs [50]. It results in in-frame insertions, usually concentrated within or following the C-helix that dictates the activation status of EGFR [51]. In glioblastoma, the most frequently (~30%) occurring EGFR mutation is *EGFRΔIII* (EGFR variant III), which results from the in-frame deletion of 801 base pairs spanning exons 2–7 of the coding sequence, resulting in ligand-independent activation of EGFR tyrosine kinase activity [52,53,54].

The *LREA* deletion of exon 19 of EGFR is shown to increase EGFR autophosphorylation and to activate downstream pathways AKT and STAT, thus promoting survival and cell growth [55]. With this mutation, the EGFR dimer exhibits increased stability as it tightens the molecular contacts of arginine ARG744 and asparagines ASP974 and ASP976 from the reciprocal monomers [56]. According to Gu and coauthors [57], this deletion is the most frequent (63.4%) in patients primarily diagnosed with NSCLC.

The 21st exon point mutation *L858R* is also a common activation mutation of *EGFR*, accounting for nearly 40% of all EGFR mutations. The *L858R* mutation locks the kinase in a constitutively active state by preventing the activation loop segment (residue 858 and flanking residues) from adopting the inactive, helical conformation, which leads to about 50-fold greater activity of the mutant EGFR [58].

*EGFRvIII* (EGFR with 2–7 exon deletion) lacks a ligand-binding domain and is constitutively active. It is the most common EGFR mutation occurring in glioblastoma [52]. *EGFRvIII* pathologic isoform does not contain amino acids 6-273 of wild-type EGFR, and it results in the formation of a new glycine residue at the junction site. This alteration imitates the effects of ligand binding and triggers changes in the receptor conformation by ultimately activating downstream signaling pathways [59].

In addition, the patients may have uncommon *EGFR* mutations. The use of next-generation sequencing (NGS) has become a novel diagnostic method for detection, which led to the identification of increasingly rare or atypical EGFR mutations. For example, *EGFR* fusion mutation *EGFR–SEPT14* was found in a patient with colorectal adenocarcinoma. The exon 24 of *EGFR* was fused to the exon 10 on *SEPT14* while retaining the EGFR tyrosine kinase domain. This tumor appeared to be sensitive to erlotinib treatment, and the patient developed a partial response following therapy [60]. Also, the same fusion oncogene was identified in a lung adenocarcinoma case [61]. Additional four fusions have been identified in lung cancer patients: *EGFR–TNS3*, *EGFR–PURB*, *EGFR–RAD51*, *KIF5B-EGFR*, and *EGFR–ZCCHC6*. Although it is problematic to catch clear-cut connections with therapy response here due to a lack of statistically sufficient patient groups, these rare mutations most probably somewhat influenced the efficacy of targeted therapy [62,63].

In addition, activating mutations of downstream genes of regulatory kinases involved in the Ras/MAPK signaling pathway, such as KRAS, NRAS, and BRAF, are exceptionally frequent and appear in more than 90% of pancreatic, ~32% of lung, and ~52% of colon cancers [64]. Ras family members activate the MAPK signaling pathway, which is originally initiated by a ligand binding to a receptor tyrosine kinase (RTK), such as the EGFR [65].

Ras proteins are downstream targets of EGFR and are normally activated upon receptor stimulation, but with mutations, they may become permanently active. Mutations of Ras family genes are mainly found in codons 12 and 13. Consequently, a patient with both EGFR sensitizing mutations and Ras mutations in the tumor may not respond to targeted therapy due to proliferative signal transduction by an active Ras oncoprotein regardless of upstream inhibition of EGFR by a drug. Thus, oncogenic mutations in Ras may serve as the markers of resistant phenotype towards EGFR-targeted treatments [66].

## 3. EGFR-Targeted Therapies

Since EGFR is frequently overexpressed and/or mutated in multiple cancer types, it has prompted the development of a number of specific targeted therapeutics. Currently, there are two classes of EGFR-specific cancer drugs: monoclonal antibodies (mAbs), which bind to the extracellular domain of the transmembrane receptor and block its dimerization, and small-molecule tyrosine kinase inhibitors (TKIs), which bind to the adenosine triphosphate (ATP) binding site [67] (Figure 3b, Table 1). In turn, TKIs can be classified according to the mechanism of binding with the receptor tyrosine kinase domain: type I (binding with ATP site in mainly active conformation), type II (binding with ATP site plus back pocket, DFG(Asp855-Gly857)-out, in inactive conformation), type I½ (binding to a DFG-in, in inactive conformation), type III inhibitors binding to allosteric sites, and type IV inhibitors which generally form covalent adducts with their target protein [68,69]. EGFR-targeted drugs are currently widespread, globally approved, and are used worldwide for hundreds of thousands of patients per year.

## 4. First Generation of EGFR-Targeted Drugs

The function of tyrosine kinase enzymes (including those of the ERBB/HER receptor family) is transferring γ-phosphate of an ATP molecule to the tyrosine residue of the substrate, thus initiating signal transmission to further downstream components [70]. Thus, targeting the tyrosine kinase activity of EGFR may abrogate its signal transducer capacity inside the cell. Inhibition of the tyrosine kinase activity by a class of organic quinazolines was described first in 1994 [71], and two years later, Wakeling and colleagues reported the pharmacological characteristics of gefitinib [72].

-Gefitinib, or ZD1839 (Iressa; Astra-Zeneca Pharmaceuticals), is an oral anilinoquinazolone with a structure formula presented in Figure 4a. By interacting with several amino acid residues, gefitinib takes up space in the ATP-binding site. The first nitrogen of the quinazoline ring creates a hydrogen bond with Met793 in the hinge region and interacts hydrophobically with Leu718, Val726, Lys745, Met766, Leu788, Thr790, and Leu844 residues [73].

In a dose-dependent manner, gefitinib could inhibit the growth of EGFR-overexpressing cell lines and human-derived tumor xenografts in mice [74]. Also, it has been shown that gefitinib also inhibits the phosphorylation of HER2 in HER2-overexpressing model cell lines and prevents the growth of BT-474 xenografts established in nude mice [75,76].

By inhibiting EGFR tyrosine phosphorylation, gefitinib affects downstream signaling cascades in the tumor cell. Many in vitro studies have been performed to determine cellular changes caused by gefitinib. For example, the addition of gefitinib to the cell growth media caused cell autophagy and apoptosis associated with the inhibition of the PI3K/Akt/mTOR signaling axis. In human lung cancer cell line PC9 with Glu746-Ala750 deletion mutation in exon 19 of EGFR, gefitinib inhibited phosphorylated Akt (p-Akt) and phosphorylated mTOR (p-mTOR) expression [77]. In A549, EGFR wild-type cell line, gefitinib also decreased the expression of PI3K, AKT, p-AKT, mTOR, and A549 cells, and apoptotic rate increased in a dose-dependent manner following gefitinib treatment [78].

Gefitinib has been shown to inhibit cell proliferation in multiple tumor cell lines. This effect was associated with cell cycle arrest in the G1 phase in many studies. In the A459 cell line, gefitinib inhibited the expression of transcription factor E2F-1, which determines the G1/S transition of the cell cycle. In cell line A431, gefitinib caused upregulation of p27, cyclin-dependent kinase inhibitor and inhibitor of cell cycle progression [79]. Also, the increased expression of p27 was observed in pancreatic cancer cells PANC-1 and CFPAC-1 that underwent gefitinib treatment [80]. In human cervical cancer cell lines HeLa and Siha, gefitinib also suppressed cell proliferation and caused G1 cell cycle phase arrest [81].

The potential antitumor effect of this drug allowed clinical trials to be launched, leading to approval by the FDA (U.S. Food and Drug Administration) in 2003 as monotherapy treatment for patients with locally advanced or metastatic non-small cell lung cancer (NSCLC) after randomized double-blind clinical trials [82]. In 2004, the correlation between NSCLC tumor sensitivity to gefitinib and mutations in the EGFR tyrosine kinase domain was discovered. It was concluded that patient screening based on the presence of EGFR-activating mutations in the tumor could help select potential responders to gefitinib therapy [83]. This is based on the rationale that EGFR-mutant tumors had demonstrated enhanced tyrosine kinase activity in response to EGF and also increased sensitivity to gefitinib [43]. On 13 July 2015, the FDA approved gefitinib for the treatment of patients with advanced or metastatic NSCLC whose tumors have EGFR exon 19 deletions or exon 21 *L858R* substitution mutations as detected by an FDA-approved test [84,85] (Table 1). Gefitinib is approved in 91 countries for the treatment of adult patients with locally advanced or metastatic EGFR NSCLC [84].

-Erlotinib. In 2004, the FDA approved another low molecular mass quinazolinamine EGFR tyrosine kinase inhibitor, erlotinib (Tarceva), as monotherapy for the treatment of patients with locally advanced or metastatic NSCLC, for whom chemotherapy treatment was ineffective [86] (Figure 4b, Table 1). Similar to gefitinib, erlotinib’s mechanism of action is based on reversible binding to the intracellular tyrosine kinase domain of the EGFR receptor and blocking the binding of ATP molecules, thus preventing further tyrosine phosphorylation activities [87].

Erlotinib has been shown to be effective on various types of cancer model cell lines. For example, erlotinib treatment of A549, EGFR wild-type cell line, significantly inhibits cell proliferation in a dose-dependent manner. Also, exposure to erlotinib increased intracellular reactive oxygen species (ROS) production and G0/G1 cell cycle arrest, thus leading to increased apoptosis [88]. In A431, another cell line with high expression of wild-type EGFR, it has been shown that erlotinib, similar to gefitinib, inhibits phosphorylation of ERK and Akt, major regulators of Ras/ERK/MAPK and PI3K/Akt/mTOR signaling axes [89].

Despite the proven efficacy of erlotinib in vitro, it has been shown that human blood serum of healthy donors can donor-specifically dramatically abolish the cell growth rate inhibition by erlotinib, and this effect correlates with a decreased activity of ERK1/2 proteins and abolishment of drug-induced G1S cell cycle transition arrest. Bioinformatic analysis revealed that EGF/human serum-mediated A431 resistance to EGFR drugs can be largely explained by the reactivation of the MAPK signaling pathway [90].

In vivo study of erlotinib’s effect in nude mice carrying EGFR-mutant xenografts derived from NSCLC cell lines HCC827, PC9, and H1975 showed that high-dose treatment improved the progression-free survival of animals in two EGFR-mutant xenografts derived from HCC827 and PC9 cell lines, both carrying activating exon 19 deletion of EGFR. However, in this study, erlotinib in regular doses proved ineffective in the H1975 cell line bearing the *T790M* mutation EGFR in vitro, yet high doses of erlotinib were capable of inhibiting tumors with *T790M*-mutant EGFR in vivo to some extent as well [91].

Erlotinib is currently approved for the treatment of NSCLC and pancreatic cancer. Specifically, it was proved to be highly effective against EGFR-positive tumors with exon 19 deletion or exon 21 *L858R* substitution in a randomized phase III trial [92]. It showed efficacy in a phase III trial comparing erlotinib with chemotherapy in advanced NSCLC patients: erlotinib significantly improved progression-free survival (PFS) and overall survival (OS) [93]. Another phase III study demonstrated that first-line erlotinib treatment of patients with EGFR mutation-positive NSCLC provides a statistically significant improvement in PFS in comparison to chemotherapy treatment with gemcitabine/cisplatin (11.0 months versus 5.5 months) [94] or docetaxel/cisplatin (9.7 months versus 5.2 months) [92]. Also, erlotinib may be considered as a first-line maintenance treatment for NSCLC patients who do not show any progression following four cycles of chemotherapy, as median PFS was significantly longer for patients treated with erlotinib than with placebo: 12.3 weeks versus 11.1 weeks [95]. Furthermore, the combination of erlotinib and chemotherapy by gemcitabine has been approved by the FDA as a first-line treatment option for individuals with locally advanced and metastatic pancreatic carcinoma. This regimen showed a statistically significant advantage over single-agent gemcitabine in a phase III study (OS median 6.24 months for erlotinib/gemcitabine vs. 5.91 months for gemcitabine alone) [96]. Also, it has been shown that the lower dose of erlotinib (100 mg/d (per day) or 254 µmol/d) achieved comparable efficacy compared with the standard dose of gefitinib (250 mg/d or 559 µmol/d) in EGFR-mutated NSCLC [97]. As for EGFR wild-type tumors, there is evidence that the use of erlotinib can also increase overall survival in NSCLC patients [98]. Despite the high efficacy of gefitinib and erlotinib for patients with activating mutations of EGFR, there is also a possibility of acquired treatment resistance, which may be due to secondary EGFR mutations. For example, the T790M missense mutation in exon 20 of EGFR that is connected with acquired treatment resistance is found in ~60% of lung adenocarcinoma patients [99]. Another proven secondary resistance mechanism is the amplification of *MET*, a proto-oncogene that encodes a different heterodimeric transmembrane tyrosine kinase receptor [100].

-Lapatinib (GW572016, Tykerb/Tyverb, GlaxoSmithKline, London, UK) is a dual EGFR/HER2 TKI that reversibly binds to the ATP-binding site of the receptor with structure formula presented in Figure 4c [101]. Lapatinib forms two hydrogen bonds with EGFR Thr790 and Lys745 from the ATP-binding pocket [102]. In 2007, the FDA approved lapatinib in combination with capecitabine for the treatment of advanced or metastatic breast cancer (Table 1) [103].

In in vitro studies, lapatinib has been shown to be an effective growth inhibitor of tumor cells overexpressing EGFR or HER2 receptors in a cell-based proliferation assay using protein staining. For example, IC_50_ for EGFR-overexpressing epidermoid carcinoma cell line A431 for lapatinib is comparable with gefitinib and erlotinib (0.16 μM for lapatinib vs. 0.08 μM and 0.1 μM for gefitinib and erlotinib, respectively). Interestingly, for BT-474, HER2-overexpressing ductal breast carcinoma cell line, IC_50_ concentration of lapatinib was 0.1 μM, but IC_50_s for erlotinib and gefitinib were 1.1 μM and 9.9 μM [104].

As for other types of cancer cell lines, lapatinib is known to inhibit cell proliferation of NB4, the cell line originating from acute promyelocytic leukemia. Twenty-four-hour lapatinib treatment induced S-phase arrest (~40–60%) in NB4. Double staining with FITC-labeled annexin-V and PI analysis revealed an increased percentage of apoptosis from ~5% in control cells to ~60% under 20 µM lapatinib treatment. Analysis of the levels of Akt, p-Akt, p38MAPK, p-p38MAPK, JNK, and p-JNK revealed that lapatinib notably downregulated the expression of p-Akt and upregulated the expression of p-p38MAPK and p-JNK, suggesting stimulation of apoptosis potentially through the p38MAPK and AKT signaling pathways [105]. In gastric cancer cell lines, response to lapatinib correlated significantly with HER2 expression: in only HER2-amplified SNU-216 and NCI-N87, 2 out of 11 cell lines were sensitive to lapatinib, and it caused G1-phase shift (~50–80%) and an increase in the expression levels of cell cycle inhibitor p27^KIP1^, as well as the downregulation of cMyc and Cyclin D [106]. An in vitro study of lapatinib effect on endometrial cancer cell lines showed that compared to the non-overexpressing cell lines, the IC_50_ values of endometrial cancer cell lines that overexpressed HER2 (USPC2, USPC1) were significantly lower (0.33 vs. 4.15 µM). When cells were treated with lapatinib, there was a gradual decrease in the phosphorylation of EGFR and HER2, as well as their downstream signaling intermediates AKT and ERK [107].

Despite demonstrated activity against HER2-positive cell lines, some cellular growth factors, such as HRG1/Neuregulin-1, have been found to have a negative effect on the action of lapatinib. In NCI-N87, the gastric carcinoma cell line, and OE19, esophageal adenocarcinoma cell line, both sensitive to lapatinib, exposure to HRG1 together with lapatinib rescued cells from lapatinib-induced G1 cell cycle arrest and apoptosis. Also, the addition of HRG1 reactivated HER3 and AKT in the presence of lapatinib in these cell lines [108].

The growth of HN5 and BT474 tumor xenografts was inhibited in a dose-dependent manner with lapatinib treatment in vivo, and complete growth inhibition was observed at higher doses [104]. An in vivo metastasis assay in mice revealed that lapatinib treatment was able to prevent metastatic outgrowth of breast cancer cells 231-BR-HER2 in the brain. Mice treated with lapatinib had fewer metastatic foci than those treated with vehicle, as determined by whole-brain imaging [109]. When HER2-amplified N87, the gastric carcinoma cell line, xenografts were treated with lapatinib and tumor regression was observed. However, the combination of lapatinib and HER2-targeted drug trastuzumab induced a near-complete tumor regression in all the mice that were treated [110].

Lapatinib, when used in combination with capecitabine, was approved by the FDA in 2007 for the treatment of HER2-positive metastatic breast cancer (MBC) in patients who have previously received therapy, including an anthracycline, a taxane, and trastuzumab. In a phase III study, OS times were 75.0 weeks for the combination of lapatinib and capecitabine treatment and 64.7 weeks for capecitabine treatment alone [111]. Lapatinib was approved by the FDA in 2010 for the treatment of hormone receptor-positive metastatic breast cancer in postmenopausal women. For patients with HR-positive, HER2-positive breast cancer, the addition of lapatinib to letrozole therapy significantly decreased the risk of disease progression compared to letrozole–placebo. The PFS for patients receiving the combination therapy was 8.2 months, while those on letrozole–placebo had a median PFS of 3.0 months [112]. In a phase III trial of lapatinib plus paclitaxel with placebo plus paclitaxel as a first-line treatment for MBC, it showed that the addition of lapatinib to paclitaxel did not provide any benefits for patients with HER2-negative metastatic breast cancer. However, HER2-positive patients who received first-line therapy with paclitaxel–lapatinib experienced improvements in clinical outcomes: OS times were 104.6 weeks for combination vs. 82.4 weeks for paclitaxel–placebo) [113].

Focused study reports indicate that patients with advanced NSCLC and TKI-sensitive EGFR mutations may become insensitive to treatment and develop progressive disease after 12 months on average [114]. In that case, there is an option of a second-line therapy: the decision to choose the appropriate second-line treatment largely depends on whether the EGFR T790M mutation is present or absent, which can be detected in either plasma or tumor tissue. If the tumor proves to be *T790M*-positive, selective EGFR TKIs designed especially for mutant EGFR, for example, osimertinib, can be an option. In case of a negative result, platinum-based chemotherapy is applied [114]. Despite the development of new generations of targeted cancer therapeutics, first-generation EGFR-targeted drugs are still used against tumors with EGFR-activating mutations or EGFR-overexpressing tumors (EGFR-positive tumors), either in combination with other drugs or alone [115,116].

## 5. Second Generation of EGFR-Targeted Drugs

Emerging cases of secondary resistance to erlotinib and gefitinib forced researchers and the industry to develop new EGFR-specific therapeutics with the potential to overcome it. Second-generation EGFR TKIs were developed to address acquired resistance by inhibiting additional partner receptor tyrosine kinases (such as HER2) or irreversibly binding to the kinase domain and thereby abrogating downstream EGFR signaling.

-Afatinib (Giotrif, BIBW2992, Figure 4d) was approved in 2013 for the treatment of metastatic NSCLC carrying activating EGFR exon 19 deletions or exon 21 *L858R* substitution [117]. Its mechanism of action is different from first-generation EGFR inhibitors erlotinib and gefitinib, as afatinib irreversibly inhibits autophosphorylation of EGFR, HER2, and HER4 receptors by forming irreversible covalent bonds with ATP-binding sites (Table 1) [118]. It has a distinct acrylate side chain that covalently binds to the EGFR C797 residue, thus resulting in irreversible inhibition of the EGFR tyrosine kinase [119].

In cell culture studies, afatinib was more effective than erlotinib, gefitinib, or lapatinib in inhibiting the survival of lung cancer cell lines harboring wild-type (H1666) or *L858R/T790M* (NCI-H1975) EGFR, with IC_50_s (half-maximal inhibitory concentration is drug concentration required for 50% inhibition) below 100 nM, whereas these cells were resistant to the first-generation drugs. In cytotoxicity assays, afatinib was also 100-fold more effective than erlotinib in Ba/F3, murine interleukin-3 dependent pro-B cells carrying mutations causing resistance to erlotinib (*T790M* or extracellular domain mutations *A289V* and *R108K*). The inhibition of autophosphorylation in EGFR-mutant (carrying *L858R*/*T790M* or *L858R* mutation) cell lines NIH-3T3 by afatinib was also ~100-fold more effective than by erlotinib when testing the dose–responses for afatinib and erlotinib on EGFR autophosphorylation on wide concentration ranges (0–10,000 nM) [120]. In a head and neck squamous cell carcinoma (SCCHN) cell culture panel, afatinib almost completely reduced the rate of cell growth, and some cell lines (SCC25, SCC58, Detroit 562) showed a higher sensitivity to afatinib compared to gefitinib, with 10–300 fold lower IC_50_ [121]. Also, afatinib could effectively arrest the growth of nasopharyngeal carcinoma cell lines in vitro and suppress EGF-induced phosphorylation of EGFR, AKT, and ERK [122].

Afatinib was also proved to be more effective against NSCLC cell lines carrying the EGFR exon 19 deletion, most probably due to more efficient inhibition of EGFR phosphorylation [123]. In xenograft models, afatinib showed strong activities in EGFR *L858R*/*T790M* or HER2-overexpressing tumors [120]. In the head and neck squamous carcinoma cell line HN5 tumor xenograft, afatinib was found to be more effective in arresting tumor xenograft growth than three other TKIs with ERBB/HER-targeting activities (lapatinib, erlotinib, and neratinib) [121].

Phase III trials showed the efficacy of afatinib as first-line therapy in comparison with chemotherapy (pemetrexed/cisplatin in “LUX-Lung 3” and gemcitabine/cisplatin in “LUX-Lung 6” trials) in patients with metastatic lung adenocarcinoma carrying EGFR mutations. The drug significantly improved the objective response rate in patients with brain metastases. In a combined analysis, PFS with afatinib was significantly improved in comparison with chemotherapy (8.2 months versus 5.4 months) [124]. Another randomized-control trial proved that, in comparison with gefitinib-only treatment, afatinib improved outcomes in EGFR mutation-positive NSCLC patients who did not receive any previous therapy [125]. Also, it was demonstrated that afatinib could be effective for patients with HER2-positive breast cancer that progressed after treatment with trastuzumab, as the drug inhibits the activity of at least three HER family receptors [126].

-Neratinib (Nerlynx, HKI-272, Figure 4e [127]) is a second-generation HER2/EGFR/HER4 TKI [128]. It covalently combines with cysteine residues Cys-773 and Cys-805 of ATP-binding domains of HER1, HER2, and HER4, thus inhibiting the receptor function [129].

The anti-proliferative effects of neratinib were examined in vitro across a panel of 115 cancer cell lines by ATPlite 1step Luminescence Assay System for analysis of cell viability. In this panel, there were 22 cell lines harboring point mutations or amplifications of the *HER2* (*n* = 9), *HER3* (*n* = 10), or *EGFR* (*n* = 10) genes, and neratinib was proven to be the effective drug with IC_50_s comparable to other TKIs in this study [130]. Interestingly, neratinib has demonstrated efficacy in cell lines derived from triple-negative breast cancer (TNBC). The IC_50_ values for neratinib varied from 0.06 µM to 1.9 µM across the 14 TNBC cell lines, and there was no correlation between IC_50_ values and levels of EGFR and HER2 [131]. Also, neratinib inhibited proliferation, dose-dependently induced G0/G1 cycle arrest, and promoted apoptosis of HL-60, the human acute myeloid leukemia cell line [132]. According to [133], the combination of trastuzumab and neratinib was found to inhibit growth in SKBR3 and BT474 cells that had developed resistance to trastuzumab in vitro.

In xenograft models overexpressing HER2 (BT474) and EGFR (SKOV-3 and A431), neratinib dose-dependently inhibited tumor growth: almost by ~70–90% in xenografts of BT474, ~30–60% in xenografts of SK-OV-3, and ~32–44% in xenografts of A431 [128]. Another study showed that a combination of trastuzumab and neratinib was additive in tumor growth inhibition in the BT474 xenograft model, leading to decreased activities of HER2, Akt, and ERK pathways [133].

In a phase II trial of advanced HER2-positive breast cancer patients with and without prior trastuzumab treatment, neratinib demonstrated considerable clinical efficiency. PFS times were 22.3 and 39.6 weeks for patients in the trastuzumab-treated and trastuzumab-naïve cohorts, respectively. The objective response rates were 24% for patients who previously received trastuzumab treatment and 56% for patients with no prior trastuzumab treatment [134]. Another phase I/II trial examined a combination of neratinib and capecitabine as a treatment option for patients who previously received HER2-targeted drugs. Patients who were not treated with lapatinib before had an objective response rate for neratinib plus capecitabine of 64%; for those who received lapatinib treatment previously, the objective response rate was 57%, with PFS of 40.3 and 35.9 weeks, respectively [135].

The FDA has granted approval for the use of neratinib as an adjuvant treatment option for early-stage HER2-positive breast cancer patients who have already undergone a one-year course of trastuzumab treatment [136]. Approval was based on the ExteNET phase III trial of neratinib following adjuvant trastuzumab treatment, as disease-free survival was 94.2% in patients treated with neratinib compared with 91.9% in those receiving placebo for 1 year after trastuzumab-based therapy [137].

In a phase III trial of neratinib plus capecitabine against lapatinib plus capecitabine in HER2-positive metastatic breast cancer patients who previously received ≥2 HER2-directed MBC regimens, the combination of neratinib plus capecitabine significantly improved PFS by 2.2 months (8.8 months versus 6.6 for lapatinib plus capecitabine). The objective response rates were 32.8% for neratinib plus capecitabine and 26.7% for lapatinib plus capecitabine, and median OS were 21 months vs. 18.7 months, respectively [138]. The FDA approved the use of the neratinib and capecitabine combination for patients with HER2-positive metastatic breast cancer who had been treated with at least two prior therapies in this context based on the promising results of the clinical trial.

-Dacomitinib (Vizimpro, PF-00299804, Figure 4f [139]), another second-generation EGFR inhibitor, was approved by the FDA in 2018 as the first-line treatment of patients with metastatic NSCLC with EGFR-activating mutations (exon 19 deletion or exon 21 substitution *L858R*) (Table 1) [140]. This drug also has activity against EGFR, HER2, and HER4 receptors, which are inhibited through irreversible covalent binding of the drug at the edge of the ATP-binding cleft of tyrosine kinase domain [141]. For EGFR, irreversible inhibition is achieved by interacting with EGFR C797, similar to afatinib [119].

In NSCLC cell lines harboring endogenous EGFR *T790M* mutation, dacomitinib proved itself as an effective agent in vitro. In cell lines with *L858R* mutation or wild-type *EGFR*, dacomitinib had 10 times lower IC_50_ (μM) than reversible EGFR inhibitor gefitinib. For example, the IC_50_ of dacomitinib in the H3255 cell line carrying *L858R* mutation was 0.007 μM versus 0.075 μM for gefitinib. In wild-type *EGFR* cell lines H1819 and Calu-3, IC_50_ values of dacomitinib were 0.029 and 0.063 μM versus 0.42 and 1.4 μM for gefitinib, respectively [142]. In NSCLC cell line H1975 with both initial EGFR-activating mutation *L858R* and first-generation drug resistance mutation *T790M*, dacomitinib effectively inhibited proliferation in contrast to gefitinib [143]. In vitro SCCHN model cell lines were sensitive to dacomitinib, and the drug caused inhibition of AKT, ERK, mTOR, and STAT3, major EGFR pathway downstream signaling molecules. Forty-eight-hour dacomitinib treatment caused a ~20% dose-dependent increase in the G0/G1 cell population in the wild-type EGFR SCCHN cell lines tested (FaDu, UT-SCC-8) [144].

Furthermore, an in vitro study of dacomitinib in human bladder cancer cell lines overexpressing ERBB/HER family proteins showed superiority of dacomitinib in comparison with cetuximab or trastuzumab: for example, dacomitinib was superior to 2 μM trastuzumab (*p* = 0.0005); 2 μM cetuximab (*p* = 0.042) for UM-UC-6, the transitional bladder carcinoma cell line. In UM-UC-6, dacomitinib inhibited the phosphorylation of EGFR, ERK, and Akt, caused G1 cell cycle arrest (with ~69–77% phase shift), and induced apoptosis [145].

In xenograft tumors obtained from the SCCHN cell line FaDu, dacomitinib reduced tumor cell proliferation and EGFR phosphorylation and caused a delay in tumor growth by 13 days in comparison to the control group [144]. In a UM-UC-6 xenograft bladder cancer model, dacomitinib proved to be a more effective agent than lapatinib. The administration of dacomitinib alone and in combination with chemotherapy resulted in significantly lower xenograft weights compared to no treatment or chemotherapy alone (~280 mg of tumor weight when mice were treated with gemcitabine–cisplatin versus almost 0 mg under dacomitinib or combination treatment). The weights of xenografts treated with chemotherapy alone were not significantly different from those that received no treatment. So, in comparison to chemotherapy-only treatment, the combination of dacomitinib plus chemotherapy treatment was superior, feasible, and safe [145].

A phase III trial of dacomitinib versus gefitinib as first-line therapy for patients with EGFR mutation-positive NSCLC demonstrated improved progression-free survival: 14.7 versus 9.2 months [146], and a 4-year update on the status of patients in this trial showed overall survival benefit from first-line treatment with dacomitinib in comparison with gefitinib: OS was 34.1 months with dacomitinib versus 27.0 months with gefitinib [147]. Interestingly, it was observed that patients with the exon 21 mutation exhibited a longer PFS to dacomitinib than patients with exon 19 deletion (5.8 vs. 4.1 months) [148]. In addition, an ongoing clinical trial is currently being conducted on the clinical benefits of dacomitinib with uncommon EGFR mutations in exons 18–21, but the results have not yet been reported [149].

Other anti-kinase inhibitors have also demonstrated activities against EGFR. *Vandetanib* (*Caprelsa*, *ZD6474*) is a multitarget inhibitor of tyrosine kinase receptors EGFR, vascular endothelial growth factor receptor-2 (VEGFR-2), and RET [150] that was approved by the FDA as a treatment against advanced medullary thyroid cancer [151]. In vitro studies showed that vandetanib is also active against EGFR-expressing cutaneous squamous carcinoma (SCC) cells (A431, DJM1) [152] and human NSCLC PC-9 cells carrying a deletion in *EGFR* exon 19 (delE746-A750) [153]. However, we found no indications that this drug can work better than other EGFR inhibitors. Instead, results of a phase III study showed that patients with advanced NSCLC after prior therapy with gefitinib or erlotinib had not demonstrated an overall survival benefit for treatment with vandetanib [154].

*Brigatinib (Alunbrig*, *AP26113)* is a multi-kinase inhibitor of ALK, ROS1, FLT3, mutant variants of FLT3, and also *T790M*-mutant EGFR [155], approved by the FDA as a treatment against anaplastic lymphoma kinase (ALK)-positive metastatic NSCLC [156]. Its activity against EGFR has been evaluated in preclinical studies. The treatment of T790M/del19 and triple-del19-mutated EGFR-expressing Ba/F3 cells with brigatinib inhibited proliferation with IC_50_ less than 100 nM both in vitro and in vivo; brigatinib also inhibited the growth of PC9 cell lines with del19 alone, of double-mutant T790M/del19, and of triple-mutant C797S/T790M/del19 PC9 cells in vitro [157]. However, brigatinib showed limited efficacy in a phase I/II study evaluating drug use against EGFR-mutated lung cancer: only two out of forty-two patients had demonstrated a partial response to the drug [158].

## 6. Third Generation of EGFR-Targeted Drugs

First-generation EGFR-targeted low molecular mass therapeutics erlotinib and gefitinib have the disadvantage of being reversible inhibitors, and they are proven to be ineffective against the secondary EGFR mutations, such as the *T790M* substitution, which has been found in over 50% of EGFR-mutant NSCLC cases with acquired resistance to EGFR inhibitors [159]. To overcome this frequent mechanism of drug resistance, third-generation drugs have been developed.

-Osimertinib (Tagrisso™, AZD9291 AstraZeneca, Figure 5a) is an irreversible orally administered, EGFR-specific TKI with strong selectivity to *EGFR*-activating mutations as well as the secondary *T790M* resistance mutation in patients with advanced NSCLC (Figure 5a, Table 1) [160]. Osimertinib’s mechanism of action is the formation of a covalent bond to the cysteine-797 residue in the EGFR ATP-binding site [161].

In preclinical studies, the drug could effectively inhibit signaling pathways and growth in the NSCLC cell lines bearing both activating EGFR mutations and *T790M*-mutant EGFR. In contrast, this drug’s effects on wild-type EGFR cell lines were relatively small in magnitude, as shown in animal models of EGFR-mutated tumors [160].

In a double-blind phase III trial, osimertinib showed higher efficacy than standard EGFR TKIs (gefitinib or erlotinib) in the first-line treatment of EGFR mutation-positive advanced NSCLC, with a similar safety profile: the median PFS was significantly longer with osimertinib than with standard EGFR TKIs (18.9 months vs. 10.2 months) [162]. Another phase III study showed that osimertinib had significantly greater efficacy than chemotherapy (platinum therapy plus pemetrexed) in patients with EGFR *T790M*-mutated advanced NSCLC who had progressed after first-line therapy with EGFR TKIs, as PFS was 8.5 months among patients with metastases to the central nervous system receiving osimertinib vs. 4.2 months receiving platinum therapy plus pemetrexed [163]. Currently, osimertinib is approved by major regulatory agencies for the treatment of *T790M*-positive cancers, which progressed after treatment with first- or second-generation EGFR TKIs [164].

Despite the strong clinical effectiveness of osimertinib, patients ultimately develop secondary resistance to this therapy as well, which presents a significant obstacle given the limited number of currently available pharmacological alternatives [161].

After 8–10 months of treatment with osimertinib, secondary resistance to the drug may occur, where resistance mechanisms may include EGFR *C797S* in addition to *T790M* mutation or even loss of *T790M* mutation. Here, *C797S* mutation impairs the covalent bond formation, as the alcohol side chain in serine is significantly less nucleophilic than the thiol side chain in the cysteine residue [165].

-Almonertinib, also known as Aumolertinib, HS-10296, or Ameile, is another low molecular mass TKI with high selectivity for EGFR-sensitizing and *T790M*-resistant mutations. Similar to osimertinib, it covalently and irreversibly binds to cysteine-797 at the ATP-binding site of the EGFR tyrosine kinase domain (Table 1). The only difference in chemical structure from omisertinib is the replacement of a cyclopropyl group on the indole nitrogen (Figure 5b) [166].

An in vitro study of NSCLC cell lines showed that almonertinib significantly inhibits H1975 (EGFR-resistant mutation: L858R/T790M mutation) and HCC827 (EGFR-sensitive mutation: E746-A750 deletion) cell colony formation rather than A549 cells (wt EGFR) and induced the apoptosis in H1975 and HCC827 cells in a dose-dependent manner rather than A549 cells [167].

In phase I-II trials, almonertinib demonstrated clinical benefits and minimal toxicity in patients with EGFR *T790M*-positive NSCLC, for whom the first generation of EGFR TKI treatment was ineffective [166,168]. In a phase III trial conducted in China, the use of almonertinib as the first-line treatment was well-tolerated and showed a significant increase in PFS: 19.3 months with almonertinib versus 9.9 months with gefitinib and duration of response (18.1 months versus 8.3 months) for advanced NSCLC patients with EGFR mutations (deletion of exon 19 or *L858R* substitution) [169]. In 2020, the China Food and Drug Administration granted approval for the use of almonertinib in treating advanced NSCLC patients with T790M-mutant EGFR who had developed resistance to first- and second-generation EGFR TKIs like gefitinib and afatinib [167].

-Lazertinib (YH25448, Leclaza) is an oral third-generation EGFR TKI developed primarily for the treatment of NSCLC (Figure 5c). It targets the EGFR molecules harboring *T790M* mutation and activating mutations such as deletion of exon 19 or L858R but is ineffective against wild-type EGFR tumors (Table 1) [170]. Similar to osimertinib, it forms a covalent bond with the C797 residue in the mutated EGFR ATP-binding site [119].

An in vitro study of lazertinib showed a higher selectivity in comparison with gefinitib against Ba/F3 cells (a murine cell line that can help overcome the challenge of obtaining patient-derived lung cancer cells that have rare driver mutations [171]) expressing different mutant EGFRs: *Del19*, *L858R*, *Del19/T790M*, *L858R/T790M*. The mean IC_50_ values for cells with mentioned mutations varied from 3.3 to 5.71 nM, similar to IC_50_ obtained for osimertinib-treated cells (3.5–4.3 nM). Also, this study demonstrated the ability of lazertinib to induce apoptosis in EGFR-mutant cell lines H1975 (carrying *L858R/T790M* mutation) and PC9 (Glu746-Ala750 exon 19 deletion) [170]. In an animal model, lazertinib demonstrated remarkable tumor regression in brain metastasis and had superior brain/plasma and tumor/brain area under the concentration–time curve values compared to osimertinib [170].

A phase I-II trial showed that lazertinib was generally well tolerated and exhibited encouraging antitumor efficacy in patients with tumors containing activating and *T790M*-resistant EGFR mutations [172]. At present, lazertinib is approved in South Korea for the treatment of locally advanced or metastatic NSCLC-carrying EGFR *T790M* mutation [173].

-Furmonertinib or Alflutinib (AST2818, Figure 5d) is another third-generation TKI inhibitor that blocks EGFR with both activating mutations and secondary mutations such as *T790M* (Table 1) [174].

Interestingly, furmonertinib showed nonlinear pharmacokinetics where its apparent clearance increases over time in a dose-dependent manner, which may be attributed to its self-induction of cytochrome P450. Furmonertinib was shown to be a strong CYP3A4 enzyme inducer that can increase the formation of active or reactive metabolites [175]. AST5902 is a metabolite of furmonertinib that is pharmacologically active and has similar antitumor activity [176]. The results of phase I/II clinical trials have indicated that furmonertinib is well-tolerated and shows notable clinical effectiveness in individuals with *T790M*-mutated NSCLC who experienced progression following gefitinib or erlotinib treatment, including patients with central nervous system metastases [176]. A phase III clinical trial of furmonertinib versus gefitinib (FURLONG, NCT03787992) in patients with locally advanced or metastatic EGFR mutation-positive NSCLC showed superior efficacy of furmonertinib compared with gefitinib as first-line therapy in the Chinese population: PFS was 20.8 months in the furmonertinib group vs. 11.1 months in the gefitinib group, along with an acceptable toxicity profile [177]. Another phase III trial (FURVENT, NCT05607550) is currently in progress, which aims to compare the safety and efficacy of furmonertinib with platinum-based chemotherapy in treatment-naïve patients with locally advanced or metastatic nonsquamous NSCLC harboring EGFR exon 20 insertion mutations [178].

-Mobocertinib (TAK-788, AP32788, Figure 5e) is another third-generation EGFR inhibitor that was developed to treat NSCLC patients with EGFR exon 20 insertions [179]. Mobocertinib selectively targets EGFRex20ins variants by interacting with the C790 gatekeeper residue in the ATP-binding pocket of EGFR through its middle pyrimidine ring while also forming an irreversible covalent bond with the C797 residue [180].

In cytotoxicity assays of Ba/F3, murine interleukin-3-dependent pro-B cells engineered to express mutant variants of EGFR, mobocertinib inhibited ex20ins EGFR variants. Oral administration of mobocertinib once daily at doses of 30–100 mg/kg resulted in tumor regression in a mouse model of ex20ins EGFR tumors [181]. This drug received accelerated approval in 2021 for the treatment of patients with locally advanced or metastatic NSCLC with EGFR ex20ins mutations based on a non-randomized, open-label, multicohort phase 1/2 study (NCT02716116) that demonstrated significant clinical benefit [182]. However, mobocertinib was not effective in vitro against engineered Ba/F3 cells containing a *C797S* EGFR mutation [183].

A current ongoing phase 3 clinical trial EXCLAIM-2 (NCT04129502) was initiated to assess the safety and efficacy of first-line mobocertinib versus platinum-based chemotherapy in patients with ex20ins EGFR locally advanced or metastatic NSCLC [184]. However, recently, the FDA and Takeda Pharmaceuticals published a mobocertinib withdrawal statement due to failure to achieve the planned results of a phase III trial [185].

EGFR TKIs were proven to be useful in treating various types of solid tumors. However, due to their broad applicability, there has been a growing incidence of TKI-induced adverse effects. While TKIs have been found to interfere with the normal functioning of many organs and cause serious adverse effects in the gastrointestinal tract, kidney, thyroid, and blood, the most severe side effects were recorded for the skin and heart [186,187,188]. These include myocardial infarction, heart failure, left ventricular dysfunction, arrhythmias, and hypertension [189].

## 7. Fourth Generation of EGFR-Targeted Drugs

Over time, patients treated with third-generation EGFR TKIs develop heterogeneous resistance to this therapy, which can be either EGFR-dependent or independent [190]. The primary cause of EGFR-dependent resistance is the emergence of a specific point mutation *C797S* in the ATP-binding cleft [191,192]. Another factor is a deletion spanning the secondary mutation *T790M* [192].

To overcome these resistance mutations, fourth-generation drugs that bind to an allosteric site of EGFR are currently being developed and undergoing preclinical evaluation [193,194]. For example, after analyzing over 2.5 million chemical compounds, two non-ATP competitive compounds were discovered, EAI001 (EGFR allosteric inhibitor-1, Figure 6a) and EAI045 (EGFR allosteric inhibitor-45, Figure 6b), that target the allosteric site of EGFR and prevent its binding to ATP [194,195].

An in vitro study of EAI045 in NSCLC H1975 and NIH-3T3 cells with *L858R/T790M* mutations showed decreased EGFR autophosphorylation. Furthermore, a study was conducted on the combined use of EAI045 with cetuximab, which was effective against both *L858R/T790M* and *L858R/T790M/C797S* EGFR-mutant Ba/F3 cells and also in vivo against *L858R/T790M* and *L858R/T790M/C797S*-mutated EGFR mouse models [195].

Another reversible non-ATP competitive allosteric inhibitor of EGFR under investigation is JBJ-04-125-02 (Figure 6c), which is active against EGFR *L858R*, *L858R/T790M* or *L858R/T790M/C797S* Ba/F3 cells, but showed fewer activities against H1975 and H3255GR mutant EGFR cell models [196]. CH7233163 (Figure 6d) is another allosteric inhibitor of EGFR. It directly interacts with the gatekeeper residue T790M in addition to the P-loop and hinge regions and binds more extensively within the ATP-binding pocket [197]. CH7233163 has demonstrated activity both in vitro and in vivo against the following EGFR mutation models: *del19/T790M/C797S*, *L858R/T790M/C797S*, *del19/T790M*, *L858R/T790M*, *del19*, and *L858R*. In contrast to the previous allosteric inhibitors presented, it was also effective against *del19* EGFR-mutant models [197]. However, there were currently no clinical trial reports that could prove the clinical efficacy of the above mentioned fourth-generation EGFR inhibitors.

BLU-945 (Figure 6e) was obtained by Blueprint Medicines by optimizing the molecules from ~25,000 compound library of designed small-molecule kinase inhibitors and showed in vitro sub-nanomolar activities against the *EGFR T790M* and *EGFR T790M/C797S* mutants. In addition, it reduced EGFR phosphorylation in Ba/F3 cells: in *L858R/T790M/C797S* mutants with IC_50_ = 3.2 nM and in *ex19del/T790M/C797S* mutants with IC_50_ = 4.0 nM. In vivo experiments showed its effectiveness in mice with ex19del/T790M/C797S EGFR-mutated NCI-H1975 and Ba/F3 xenografts.

In an osimertinib-resistant EGFR *ex19del/T790M/C797S* mouse model derived from a patient with an NSCLC tumor who progressed after treatment with gefitinib and osimertinib, the tumor growth was inhibited by BLU-945 [198]. The phase I/II SYMPHONY trial (BLU-945-1101; NCT04862780) is currently ongoing to evaluate the antitumor activity, tolerability, and safety of this compound as a monotherapy and in combination with osimertinib in patients with EGFR-mutated NSCLC [199]. The initial reports showed that BLU-945 alone or in combination with osimertinib demonstrated early signals of clinical activity and was well tolerated in heavily pretreated EGFR-mutant NSCLC patients [200].

BLU-701 is another compound developed by Blueprint Medicines that is a highly selective and potent inhibitor of EGFR with *ex19del-* or *L858R*-activating mutations and the *C797S* resistance mutation with nanomolar IC_50_ (~3.3 nM) [201,202]. At tolerated doses, oral administration of BLU-701 in mice led to significant and sustained regression of the PC9 ex19del tumor xenografts [201]. The safety and effectiveness of BLU-701 in patients with EGFR-mutated NSCLC who have received previous treatment with EGFR TKIs is currently being assessed in the phase I/II HARMONY trial (NCT05153408) [203].

For two other fourth-generation EGFR inhibitors, JIN-A02 and BBT-176, the successful application in both in vitro and in vivo studies were published: JIN-A02 inhibited *ex19del/T790M/C797S* and *L858R/T790M/C797S* EGFR-mutant Ba/F3 cells (IC_50_ = 51.0 and 49.2 nM, respectively) and resulted in tumor regression in *ex19del/T790M/C797S* Ba/F3 xenograft mouse models [204]. The IC_50_ values of BBT-176 for Ba/F3 cells engineered to express EGFR *19Del/C797S*, EGFR *19Del/T790M/C797S*, and EGFR *L858R/C797S* and *L858R/T790M/C797S* were 42, 49, 183, and 202 nM, respectively. Moreover, when BBT-176 was used in combination with cetuximab, it showed a synergistic effect against the EGFR *19Del/T790M/C797S*-expressing cells. It also inhibited tumor growth in vivo in *a* patient-derived *19Del/T790M/C797S* EGFR model and induced tumor regression in a LU1235 EGFR *19Del* model.

Clinical phase I/II trials for these compounds are currently underway (NCT05394831 and NCT04820023, respectively). Additionally, treatment with BBT-176 caused a reduction in tumor size in two patients belonging to the 320 mg once-daily and 480 mg once-daily groups after receiving BBT-176 treatment for 6 and 12 weeks, respectively [205].

Finally, it should be added that new fourth-generation drug candidates are emerging permanently, being representatives of different chemical classes with various mechanisms of action [206].

## 8. EGFR-Specific Therapeutic Monoclonal Antibodies

Soon after the discovery of the EGFR receptor in the 1980s, prof. John Mendelsohn noted that the addition of EGF, the ligand of the EGFR receptor, had a negative effect on the survival of the A431 tumor cell line, which contained large amounts of EGFR. The idea was that it was possible to stop EGFR-overexpressing tumor proliferation through interference with the EGFR signaling [207]. Because EGFR permeates the cell membrane, the idea arose that monoclonal antibodies could be an effective therapeutic against tumors with increased expression of this receptor. EGFR-specific mAbs function similarly by disrupting pro-tumor growth and survival signaling through binding to growth factor receptors, thus altering their activation state or preventing ligand binding [208]. In addition, the specific binding of antibodies can recruit immune cells to recognize and target tumor cells [209]. The second pathway is indirect and acts by an antibody-dependent cellular cytotoxicity (ADCC) mechanism. For example, natural killer (NK) cells can target mAb-treated HER2/neu-overexpressing cells through the ADCC mechanism [210]. As for EGFR-targeted antibodies, cetuximab proved to have ADCC activity against tumor cells, which was dose-dependent on cell surface EGFR expression [211].

*-Cetuximab* (Erbitux, Merck Serono) was the first monoclonal antibody targeting the EGFR receptor, a human–mouse chimeric anti-EGFR mAb with the human IgG1 constant region [212]. It exhibits a strong affinity for human EGFR and effectively hinders ligand binding, ultimately resulting in the suppression of receptor phosphorylation and downstream signaling pathways [213]. The primary effect of cetuximab binding to EGFR is steric blockage of ligand access to the binding site in domain III of the receptor (Figure 3b, Table 1). The ligand binding site in domain I is not affected by cetuximab, but EGF interacts with both domains I and III to bind with high affinity and activate the receptor; therefore, effective blocking of either domain is sufficient to inactivate EGFR [214]. In addition to competitive inhibition, cetuximab binding with EGFR may also induce internalization and degradation of the receptor [215].

In vitro studies showed that, in contrast to gefitinib, cetuximab inhibited proliferation in EGFR wild-type NSCLC cell lines only but was not effective against cell lines carrying activating EGFR mutations and could not inhibit the phosphorylation of mutant EGFR [216]. In other cancer cell types, cetuximab proved to be effective in KRAS wild-type, EGFR-expressing gastric cancer cell line NCI-N87 and xenografts [217]. Although, in some studied cases, treatment with cetuximab alone had no significant effect on cell growth rate in vitro, in combination with other drugs, it could show a synergistic effect in cells originating from different cancer types [218].

It was found that the inhibition of cell growth induced by blocking EGFR activation of cetuximab deals with the induction of cell cycle arrest and apoptosis. Cetuximab induced cell accumulation in the G1 phase and increased the expression levels of cell cycle inhibitors p27^KIP1^ and p15^INK4B^ in human oral squamous cell carcinoma cell lines [219]. A similar effect was observed in SCCHN cell lines: cetuximab treatment decreased cell motility and enhanced cell arrest in the G1 phase; also, the accumulation of p27^KIP1^ was observed following cetuximab treatment [220]. In model NSCLC cell cultures, exposure to cetuximab led to an increase in proapoptotic protein Bax and a decrease in the negative regulator of apoptosis Bcl2. Thus, the increase of the Bax/Bcl2 ratio in cells following cetuximab treatment most likely relates to the initiation of apoptosis [221].

The effectiveness of cetuximab was modulated by the addition of human blood serum of healthy donors to the growth medium in vitro. The addition of 5% human blood serum to cells contributed to a decrease in the antiproliferative activity of cetuximab in the EGFR-overexpressing A431 cell line [222], and this effect correlated with a decreased activity of ERK1/2 proteins and repression of cetuximab-induced G1S cell cycle transition arrest. The expression of 75% differently expressed genes, obtained by RNA sequencing, restores to the no-drug level when human serum is added along with cetuximab. The analysis of molecular pathways revealed that the addition of human serum reactivated MAPK signaling pathways inhibited by cetuximab alone [90]. Interestingly, human serum also modulated the effect of the HER2 monoclonal antibody trastuzumab on the HER2-overexpressing cell line BT-474 [223].

Results of a phase II trial of cetuximab serving as a single agent in patients with chemotherapy–refractory EGFR-overexpressing colorectal cancer showed it is well tolerated but has modest activity [224]. When utilized in combination with chemotherapy to enhance overall effectiveness, the phase III study showed that for metastatic colorectal cancer (mCRC) patients who did not respond to previous first-line fluoropyrimidine and oxaliplatin treatments, cetuximab in combination with irinotecan improved response rate (16.4% vs. 4.2% for irinotecan) and PFS (median, 4.0 months for cetuximab with irinotecan vs. 2.6 months for irinotecan) [225]. Another phase III trial evaluated cetuximab in combination with FOLFOX as an effective standard-of-care first-line treatment regimen for patients with RAS wild-type metastatic colorectal cancer [226]. For patients with recurrent and/or metastatic SCCHN, the addition of cetuximab to standard cisplatin/carboplatin and 5-fluorouracil scheme showed an improved median PFS (5.5 months with cetuximab vs. 4.2 months without cetuximab) and median OS (11.1 months vs. 8.9 months) in a Chinese phase III trial [227]. Furthermore, for high-dose radiation therapy, it was shown that its co-administering with cetuximab is an effective approach for managing locoregionally advanced head and neck cancer, as it enhances locoregional control (24.4 months for patients treated with cetuximab plus radiotherapy vs. 14.9 months for patients treated with radiotherapy) and minimizes mortality rates, while not causing any additional adverse effects typically associated with radiotherapy for SCCHN tumors [228].

Cetuximab has been approved by the European Medicines Agency and the FDA for the use in patients with locally advanced SCCHN and in combination with irinotecan for the treatment of mCRC. In the US, cetuximab has also been approved as monotherapy for patients with recurrent or metastatic SCCHN and in patients with mCRC who cannot tolerate irinotecan-based regimens [229,230]

However, several retrospective randomized studies provided evidence indicating that administering mAbs-targeting EGFR, such as cetuximab, does not provide any benefits to patients with advanced colorectal cancer with tumors containing activating mutations in the *KRAS* and other RAS family genes [231,232].

-*Panitumumab* (Vectibix, Amgen, Inc., ABX-EGF) is a human monoclonal antibody specifically targeted at EGFR. It has the same presumed mechanism of action as cetuximab, i.e., binding to extracellular domain III of the EGFR molecule and preventing it from activating through interaction with ligands (Table 1) [233]. However, panitumumab has a higher affinity for binding with EGFR than cetuximab [234].

It has been shown that panitumumab completely prevents the formation of human epidermoid carcinoma A431 xenografts in athymic nude mice. The drug also caused tumors to completely dissolve in mice, even without additional treatment with chemotherapeutic agents or radiation [235]. Treatment with panitumumab also inhibited tumor growth in other human tumor xenografts with breast, renal, pancreatic, ovarian, and prostate tissues of origin [236].

In a phase III clinical trial, panitumumab significantly improved progression-free survival (8 weeks for panitumumab plus best supportive care vs. 7.3 weeks for best supportive care) in patients with advanced colorectal cancer that had progressed after standard chemotherapy and was also well tolerated [237]. Panitumumab in combination with infusion fluorouracil, leucovorin, and oxaliplatin (FOLFOX4) is proven to be more effective than FOLFOX4 treatment alone in mCRC patients with KRAS wild-type tumors as there was an improvement in PFS (9.6 months with FOLFOX4 plus panitumumab versus 8.0 with FOLFOX4 alone) and OS (23.9 vs. 19.7 months, respectively) [238].

In metastatic colorectal cancer patients, panitumumab is currently approved for the treatment of Ras wild-type tumors in combination with FOLFOX or FOLFIRI (5-fluorouracil, leucovorin, and irinotecan) as first-line therapy; and as monotherapy after non-effective chemotherapy treatment [239].

-Necitumumab (Portrazza, IMC-11F8) is another human monoclonal antibody against EGFR with the same mechanism of action as the previous mAbs considered in this review (Table 1) [240].

The drug effectively inhibited the growth of tumor cell lines of epidermal, pancreatic, and colorectal origins with EGFR overexpression in vitro [241]. Additionally, necitumumab has significant antitumor activity in various human xenograft tumor models and can enhance the antitumor effects of irinotecan and oxaliplatin in colorectal cancer models [242]. In vivo, necitumumab significantly reduced the expression of *CCND1* (cyclin D1) in the NCI-H1650 model. Necitumumab alone, as well as in combination with cisplatin/gemcitabine, significantly inhibited tumor growth of NSCLC models [243].

A phase III trial of necitumumab in combination with gemcitabine and cisplatin as first-line therapy improved overall survival in comparison with only gemcitabine and cisplatin in patients with stage IV NSCLC (11.5 months vs. 9.9 months) [244]. Necitumumab then received approval in the USA for the use in combination with cisplatin plus gemcitabine for the first-line treatment of squamous NSCLC [245].

## 9. Future Directions and Conclusions

Despite the obvious worldwide success of targeted drug applications against different types of cancer, there are still challenges that have not been adequately answered. Each of the next generations of EGFR-targeting drugs was designed to overcome drug resistance of tumor to the previous generation or to reach those patients for whom the previous generation did not initially work [246]. However, when the drugs are designed to target tumors with specific mutations, they become no longer effective against wild-type EGFR, thus leading to the abandonment of their use in a large patient population. Furthermore, secondary resistance can occur against the third generation of drugs (for example, the substitution of cysteine by serine at codon 797 in the ATP-binding site of the EGFR tyrosine kinase domain—*C797S* [247]). It is already forcing researchers to use combinatorial chemistry strategies and test fourth-generation EGFR-targeting candidate molecules. This ongoing dynamic resembles an arms race, wherein mutations arise, targeted therapies are developed and applied, and, in response, subsequent secondary mutations of EGFR appear in the tumor clones. The temporal benefit for the patient can span from several months to several years; nevertheless, unfortunately, the tumor inevitably can develop resistance to the targeted drugs used.

Therefore, alongside the pursuit of novel drugs that target resistance mutations, the strategic advancement of combination therapeutic protocols seems to be a reasonable route to take. This approach involves the integration of multiple drugs with distinct specificities alongside conventional chemotherapy. Moreover, additional efforts are needed to explore and implement methodologies aimed at enhancing tumor susceptibility to immunotherapy, which has been recognized as holding unprecedented promise in conferring survival advantages to susceptible patients.

An alternative area of investigation for potential solutions to EGFR inhibition resistance is the activation of alternative receptor tyrosine kinase pathways, which may circumvent or evade the EGFR signaling inhibition. Comparison of gene expression in different tumor samples that do or do not respond to treatment with targeted drugs reveals molecular pathways important for successful therapy [248]. This also opens up the possibility of utilizing combinations of therapeutic agents to act on multiple molecular targets simultaneously in order to inhibit cancer growth [249].

Thus, the current scientific problems for the therapy of EGFR-overexpressing tumors deal with the finding of molecular markers associated with tumor sensitivity to the treatment. The fact of EGFR-positivity (overexpression of the receptor and/or amplification of the gene encoding it), in principle, allows the narrowing down of the population to be treated to a significant extent but has insufficient predictive power. For example, unlike the EGFR gene expression level itself, which is a poor predictor of cetuximab treatment response in colorectal cancer [250], a broader view of the quantitively measured activation of relevant molecular pathways has shown promising results [251]. Thus, exploring the underlying cellular and molecular mechanisms that cause resistance to EGFR inhibitor treatments and utilizing personalized predictive approaches can reveal innovative strategies to improve the efficacy of EGFR-targeted therapies.

## Figures and Tables

**Figure 1 cells-13-00047-f001:**
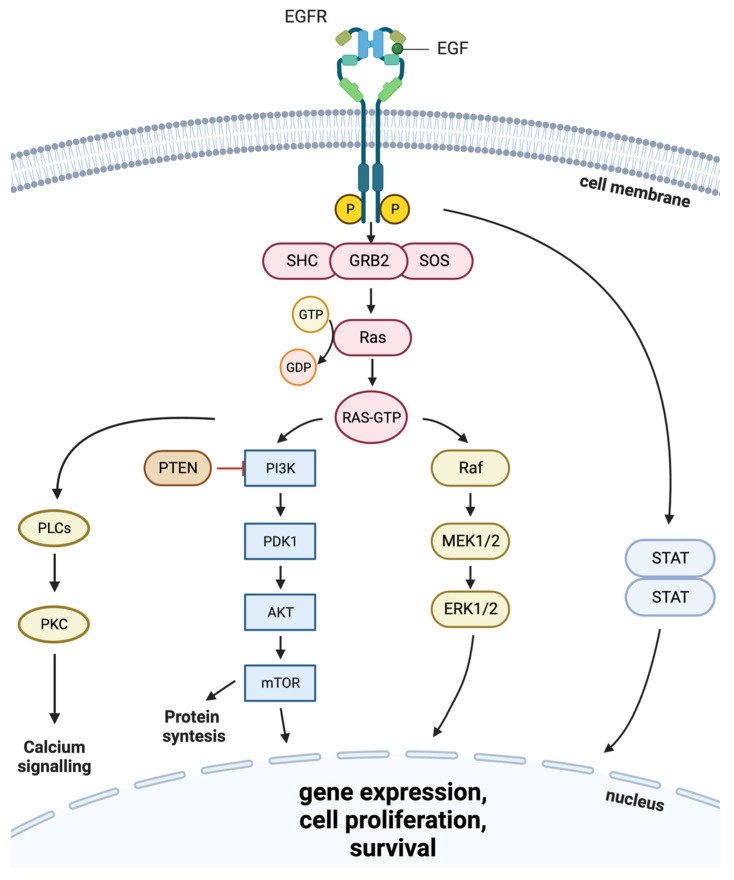
Intracellular signaling involving EGFR. The major regulatory pathways downstream of EGFR and other HER receptors are shown. Binding of specific ligands (e.g., EGF) leads to homo- or heterodimerization of receptors, thus resulting in conformational changes in the intracellular kinase domain, which results in phosphorylation and activation of the receptor. The signaling axes RAS-RAF-MEK-ERK and PI3K-AKT-mTOR, in turn, activate various downstream signaling pathways, thus leading to enhanced cell proliferation and survival. Created with BioRender.com (accessed on 1 November 2023).

**Figure 2 cells-13-00047-f002:**
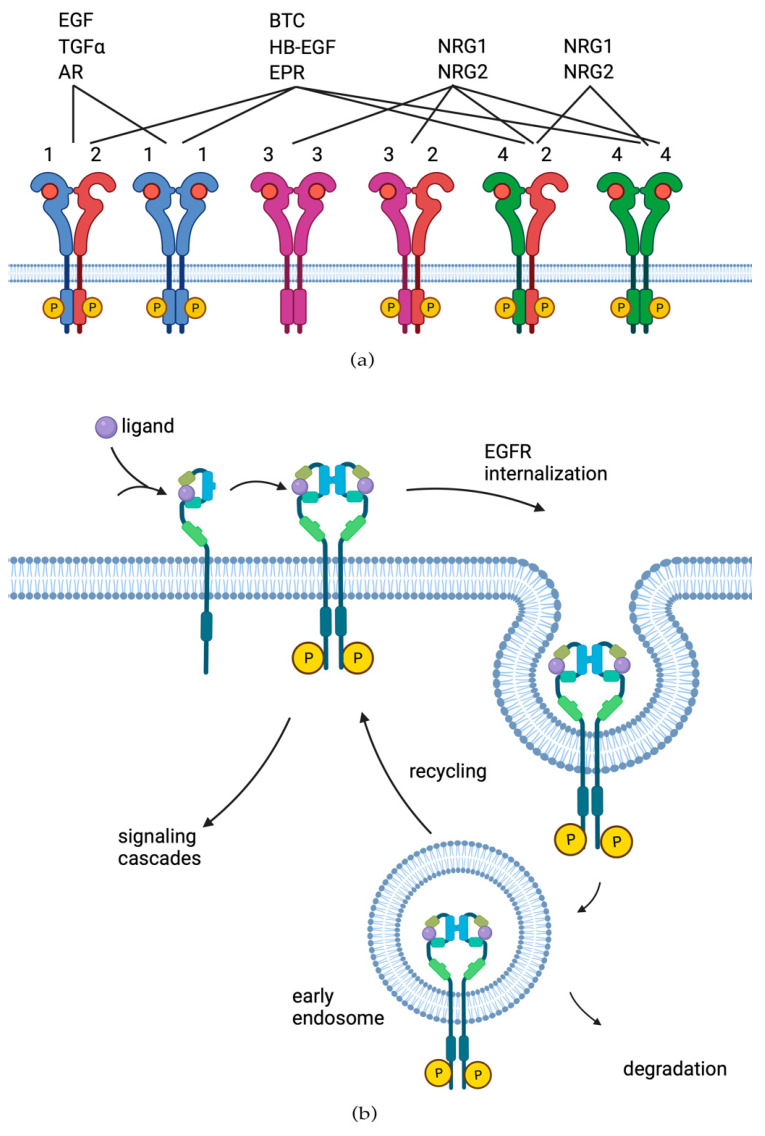
(**a**) Ligands that bind to common types of homo- and heterodimers formed by HER receptors. The following designations were used: 1—EGFR, 2—HER2, 3—HER3, and 4—HER4. (**b**) Dimerization, activation, and internalization of the EGFR receptor. Created with BioRender.com (accessed on 18 October 2023).

**Figure 3 cells-13-00047-f003:**
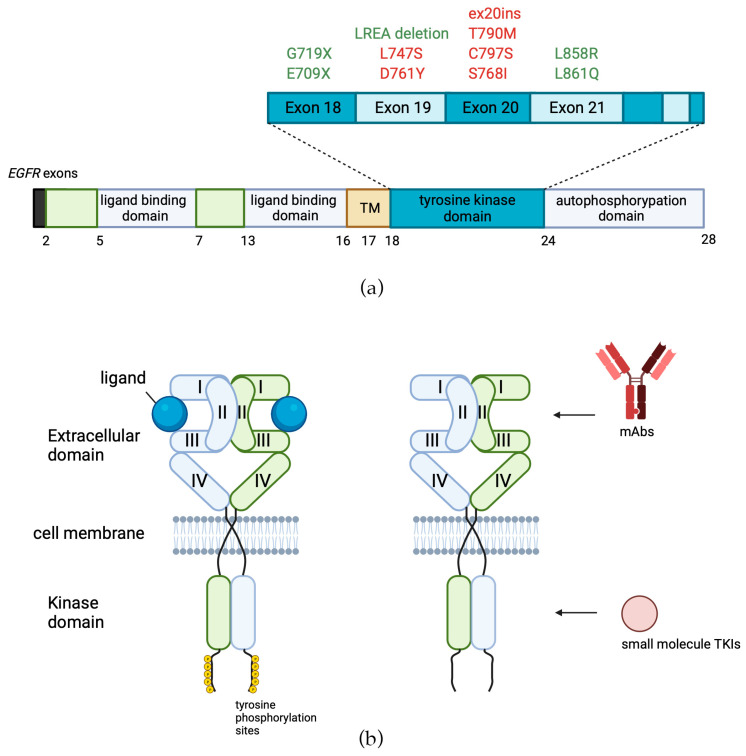
(**a**) Structure of EGFR gene. EGFR exons 18–21 encode the tyrosine kinase domain and may contain mutations, playing a crucial role in the development and progression of different cancers with a strong proven relationship to resistance (red) and sensitivity (green) to specific TKIs. (**b**) Domain view of EGFR protein. Left, a schematic diagram of ligand-bound dimerized EGFR. Right, sites of inhibition of EGFR activity by different targeted drugs (mAb: monoclonal antibodies; TKIs: tyrosine kinase inhibitors). Created with BioRender.com (accessed on 18 October 2023).

**Figure 4 cells-13-00047-f004:**
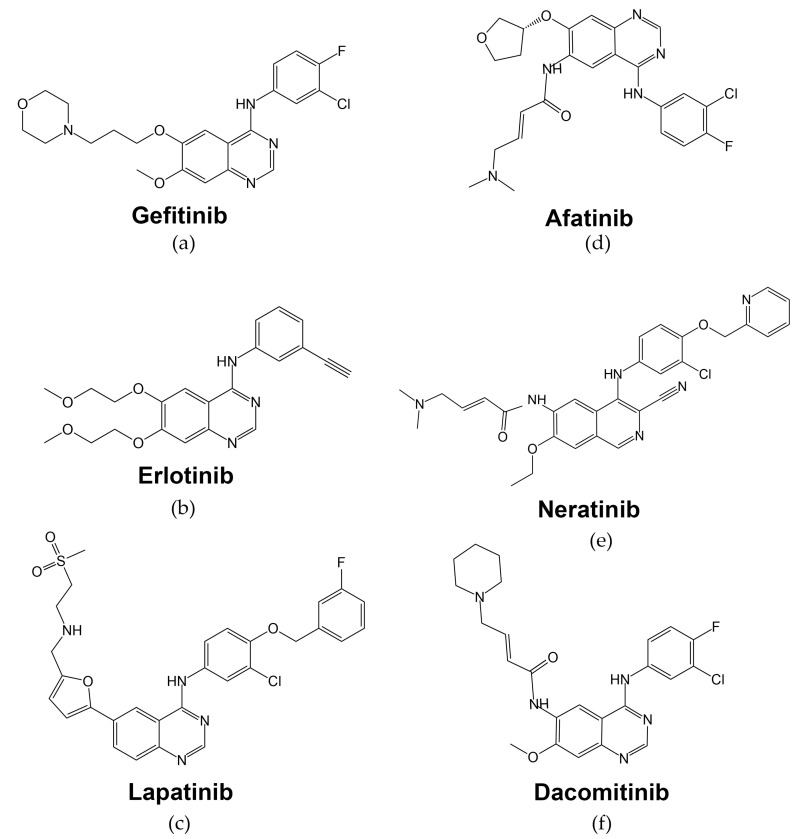
Molecular structures of members of the first and second generations of low molecular mass EGFR tyrosine kinase inhibitors.

**Figure 5 cells-13-00047-f005:**
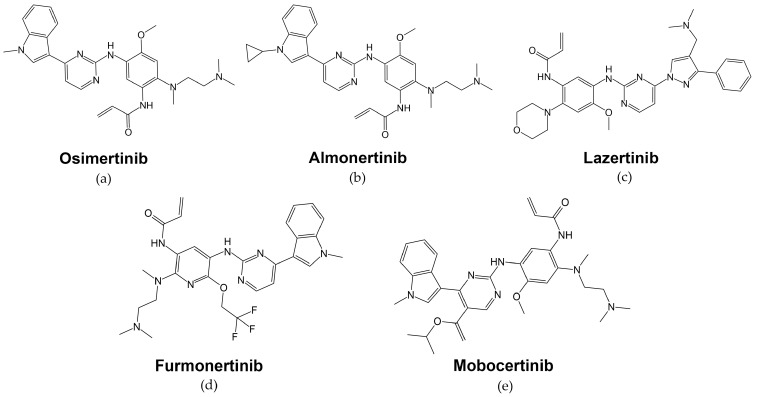
Molecular structures of members of the third-generation family of low molecular mass EGFR tyrosine kinase inhibitors.

**Figure 6 cells-13-00047-f006:**
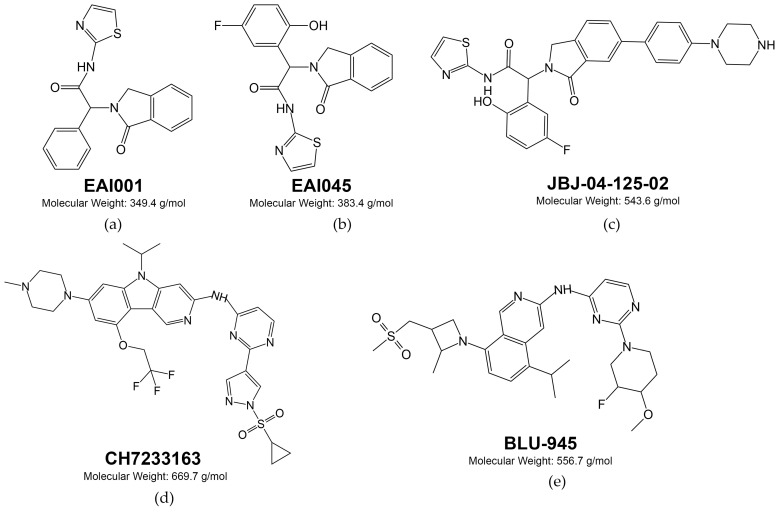
Structures of candidate molecules for the fourth-generation family of low molecular mass EGFR tyrosine kinase inhibitors.

**Table 1 cells-13-00047-t001:** Characterization of EGFR-targeting inhibitors.

Tyrosine Kinase Inhibitors					
Drug	Tumor Type	Therapeutic Indication	Molecular Target	Inhibitor Type	Molecular Markers of Efficiency
	*First Generation*				
*Gefitinib*	Advanced or metastatic NSCLC	First-line therapy for NSCLC carrying EGFR-activating mutations	EGFR: ATP-binding site	I	Activating mutations of *EGFR*: Exon 19 deletions; *L858R*
*Erlotinib*	Advanced or metastatic NSCLC, pancreatic cancer	First-line therapy for NSCLC carrying EGFR-activating mutations With gemcitabine: first-line treatment option for patients with locally advanced and metastatic pancreatic carcinoma	EGFR: ATP-binding site	I	Activating mutations of *EGFR*: Exon 19 deletions; *L858R*
*Lapatinib*	Metastatic breast cancer	With capecitabine: the treatment of HER2-positive MBC in patients who have previously received therapy (anthracycline, a taxane, trastuzumab) With letrozole: the treatment of postmenopausal women with hormone receptor positive MBC that overexpresses the HER2 receptor for whom hormonal therapy is indicated	ATP-binding site of EGFR and HER2	I½	HER2-positive status of tumor
	*Second Generation*				
*Afatinib*	Metastatic NSCLC	First-line therapy for metastatic NSCLC carrying EGFR-activating mutations	ATP-binding site of EGFR, HER2, and HER4	IV	Activating mutations of *EGFR*: Exon 19 deletions; *L858R*
*Neratinib*	Breast cancer	Extended adjuvant treatment of patients with early stage HER2-positive breast cancer, to follow adjuvant trastuzumab based therapy With capecitabine: the treatment of patients with advanced or metastatic HER2-positive BC who have received two or more prior anti-HER2 based regimens in the metastatic setting	ATP-binding site of EGFR, HER2, and HER4	IV	HER2-positive status of tumor
*Dacomitinib*	Metastatic NSCLC	First-line therapy for metastatic NSCLC carrying EGFR-activating mutations	ATP-binding site of EGFR, HER2, and HER4	IV	Activating mutations of *EGFR*: Exon 19 deletions; *L858R*
	*Third Generation*				
*Osimertinib*	Advanced or metastatic NSCLC	Adjuvant and first-line therapy for metastatic NSCLC carrying EGFR-activating mutations The treatment of adult patients with metastatic EGFR *T790M* mutation-positive NSCLC, whose disease has progressed on or after EGFR TKI therapy	ATP-binding site of the EGFR	IV	Activating mutations of *EGFR*: Exon 19 deletions; *L858R* The secondary *T790M* resistance mutation
*Almonertinib*	Advanced NSCLC	Adjuvant therapy for advanced NSCLC patients with *T790M*-mutant EGFR who had developed resistance to first- and second-generation EGFR TKIs like gefitinib and afatinib	ATP-binding site of the EGFR	IV	Activating mutations of *EGFR*: Exon 19 deletions; *L858R* The secondary *T790M* resistance mutation
*Lazertinib*	Advanced NSCLC	Treatment of locally advanced or metastatic NSCLC carrying EGFR *T790M* mutation	ATP-binding site of the EGFR	IV	Activating mutations of *EGFR*: Exon 19 deletions; *L858R* The secondary *T790M* resistance mutation
*Furmonertinib*	Locally advanced or metastatic NSCLC	Treatment of locally advanced or metastatic EGFR *T790M+* NSCLC that developed after progression on treatment with first-generation EGFR TKIs	ATP-binding site of the EGFR		The secondary *T790M* resistance mutation
**Monoclonal Antibodies**					
**Drug**	**Tumor Type**	**Therapeutic Indication**	**Molecular Target**		**Molecular Markers of Efficiency**
*Cetuximab*	Advanced or metastatic SCCHN, metastatic CRC	With radiation therapy: treatment of locally or regionally advanced SCCHN With platinum-based therapy with fluorouracil: metastatic SCCHN Metastatic SCCHN progressing after platinum-based therapy With FOLFIRI: first-line treatment of KRASwt EGFR-overexpressing mCRC With irinotecan in patients who are refractory to irinotecan-based chemotherapy: treatment of KRASwt EGFR-overexpressing mCRC; as a single-agent in patients who have failed oxaliplatin-and irinotecan-based chemotherapy or who are intolerant to irinotecan	The binding site in domain III of EGFR		KRAS wild-type status of EGFR-overexpressing tumor
*Panitumumab*	Metastatic CRC	Single agent treatment of metastatic CRC with disease progression on or following fluoropyrimidine, oxaliplatin, and irinotecan chemotherapy regimens	The binding site in domain III of EGFR		RAS wild-type status of EGFR-overexpressing tumor
*Necitumumab*	Metastatic NSCLC	With gemcitabine and cisplatin: first-line treatment of patients with metastatic NSCLC	The binding site in domain III of EGFR		EGFR-overexpressing status of tumor

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
