# Peer review of "Targeted Inhibitors of EGFR: Structure, Biology, Biomarkers, and Clinical Applications"

_cells, 2023, doi:10.3390/cells13010047_

Round 1
Reviewer 1 Report (Previous Reviewer 2)
Comments and Suggestions for Authors
I am satisfied that my specific concerns regarding parts of the previous version of the manuscript have been addressed and I thank the authors for their work.
I am still concerned that the broad but relatively shallow focus of the review will limit its appeal to students and beginners in the field, but it could be argued that there is merit also in catering to a less-experienced audience.
In this respect, with the addition of Gen 4 drugs and the new material added to Gen 3, the new manuscript would probably benefit from losing the monoclonal antibodies altogether and focusing exclusively on the TKIs.
The section on mAbs now makes a rather small part of the entire work, and its inclusion also feels a bit like an afterthought. Eliminating it would make the review shorter, easier to read and also more focused.
I know this might feel overly harsh and that you might be reluctant in undoing any of your hard work, but I ask that you consider it and see how it might be beneficial.
Apart from that, Class IV is given as VI at line 195 and in the Afatinib line of Table 1. Please amend it.
Comments on the Quality of English Language
only a few typos left, nothing major.
Author Response
Please see the attachment.

Reviewer 2 Report (Previous Reviewer 1)
Comments and Suggestions for Authors
The necessary corrections have been added to the manuscript. Thus, I conclude that the manuscript can be published in the journal.
Author Response
We thank the Referee for approving our manuscript for publication.
Reviewer 3 Report (New Reviewer)
Comments and Suggestions for Authors
In this review, authors summarized the underlying mechanisms of resistance and available personalized predictive approaches that may lead to improved efficacy of EGFR-targeted therapies. They also discussed recent developments and the use of specific therapeutic strategies such as multi-targeting agents and combination therapies, for overcoming cancer resistance to EGFR- specific drugs. This paper looks beneficial for the researchers, who work at this area and it also well-presented. It can be accepted after minor revision.
Here are some points:
· There is no need that erlotinib; cetuximab; gefitinib; panitumumab; lapatinib; osimertinib; afatinib; neratinib are included in keywords. A general phase can be chosen.
· In Figure 1, HER2, HER3 and HER4 can be also inserted and mentioned.
· If Biorender is used for Figure 2 too, it must be indicated.
· In Figure 3, there are many small molecule TKIs. It can be changed as small molecule EGFR TKIs.
· There are also reports for the effects of EGFR on melanoma. It can be also added and the role of EGFR in melanoma.
· There are dislocations in the rows of Table 1.
· The molecular weight, chemical names and the trade names of compounds are unnecessary in Figures 4 and 5, just the generic names: erlotinib, afatinib…… are adequate.
· The chemical structure of JBJ-04-125-02 must be corrected in Figure 6.
· Figures must be inserted after mentioned in the text.
· The side effects of EGFR TKIs can be mentioned.
· Future directions and conclusions part was written so general. It must be rewritten.
Comments on the Quality of English LanguageThere are many grammatical errors that authors should correct.
Author Response
Please see the attachment.

This manuscript is a resubmission of an earlier submission. The following is a list of the peer review reports and author responses from that submission.
Round 1
Reviewer 1 Report
Comments and Suggestions for Authors
The manuscript entitled „Targeted Inhibitors of EGFR: Structure, Biology, Biomarkers 2 and Clinical Applications” by N. Shaban et al. contains a review on the EGFR inhibitors. Considering the drugs currently used in the therapy of cancers, the review is quite comprehensive and provides insight into the mechanism of EGFR inhibitory activity. Moreover, the manuscript gives details on the therapeutic use of the above inhibitors in specific cancers.
However, the review could be improved significantly if the authors could address the problems I express below.
1. Why did the review omit a couple of EGFR inhibitors that had been approved for cancer treatment, e.g. Vandetanib, Brigatinib, Tucatinib, and Mobocertinib? The latter inhibitor is an interesting case as Takeda has announced plans to withdraw the drug just two years after its approval for the treatment of locally advanced or metastatic non–small cell lung cancer.
2. The interaction of the inhibitors with the amino acids present in the ATP pocket is treated cursorily by the authors. However, it would be interesting to know whether the inhibitors containing aminopyrimidine or aminoquinazoline moiety interact in a similar or different manner with these amino acids.
3. It is a pity that the authors did not provide any information on emerging four generation selective EGFR inhibitors, some of which are devoid of the typical structural elements, i.e. pyrimidine or quinazoline rings (see: M. A. Mansour et al. RSC Adv., 2023, 13, 18825).
4. Although the manuscript is readable, it contains many minor errors that need correction. Some of them I marked in the attached manuscript.

Comments on the Quality of English LanguageAlthough the manuscript is readable, it contains many minor errors that need correction. Some of them I marked in the I marked in the attached manuscript.
Reviewer 2 Report
Comments and Suggestions for Authors
Please see attached Report

Comments on the Quality of English LanguageMinor edits to the structure of some sentences and to the concordance of verbal tenses in parts of the text.